# Acquiring Iron-Reducing Enrichment Cultures: Environments, Methods and Quality Assessments

**DOI:** 10.3390/microorganisms11020448

**Published:** 2023-02-10

**Authors:** Aline Figueiredo Cardoso, Rayara do Socorro Souza da Silva, Isabelle Gonçalves de Oliveira Prado, José Augusto Pires Bitencourt, Markus Gastauer

**Affiliations:** Instituto Tecnológico Vale, R. Boaventura da Silva, 955, Bairro Nazaré, Belém 66055-090, PA, Brazil

**Keywords:** biocementation, *canga*, iron cycling, Carajás National Forest, mine land restoration, sustainable mining

## Abstract

Lateritic duricrusts cover iron ore deposits and form spatially restricted, unique *canga* ecosystems endangered by mining. Iron cycling, i.e., the dissolution and subsequent precipitation of iron, is able to restitute *canga* duricrusts, generating new habitats for endangered biota in post-mining landscapes. As iron-reducing bacteria can accelerate this iron cycling, we aim to retrieve microbial enrichment cultures suitable to mediate the large-scale restoration of *cangas*. For that, we collected water and sediment samples from the Carajás National Forest and cultivated the iron-reducing microorganisms therein using a specific medium. We measured the potential to reduce iron using ferrozine assays, growth rate and metabolic activity. Six out of seven enrichment cultures effectively reduced iron, showing that different environments harbor iron-reducing bacteria. The most promising enrichment cultures were obtained from environments with repeated flooding and drying cycles, i.e., periodically inundated grasslands and a plateau of an iron mining waste pile characterized by frequent soaking. Selected enrichment cultures contained iron-reducing and fermenting bacteria, such as *Serratia* and *Enterobacter*. We found higher iron-reducing potential in enrichment cultures with a higher cell density and microorganism diversity. The obtained enrichment cultures should be tested for *canga* restoration to generate benefits for biodiversity and contribute to more sustainable iron mining in the region.

## 1. Introduction

Detrital lateritic crusts, known as *cangas*, cover iron ore deposits in the Serra dos Carajás, Eastern Amazon, Brazil [1]. These crusts are composed mainly of iron oxyhydroxide minerals (hematite and goethite), and their genesis is strongly influenced by climatic and topographical factors, the source rock and associated vegetation and microbiota [2]. Distinct from forests on more profound soils, *cangas* are covered by rupestrian ecosystems, forming patches of open vegetation in the higher regions of the landscape [3,4]. These formations support a unique vegetation [5,6] containing 38 plant species of edaphic endemism [7] endangered by mining [8]. The restoration of lateritic crusts in the post-mining landscape may be a viable strategy to maintain this biological heritage in the region, reconciling the conservation of biodiversity with the sustainable exploitation of natural resources in one of the most important mineral provinces of the world [8,9].

Geochemical and geochronological evidence indicates that the biogeochemical cycling of iron oxide minerals was and is responsible for the formation of the *cangas* [10]. For that, in the first stage, Fe(III) is reduced to Fe(II) under anoxic conditions, and Fe(III) (oxyhydr)oxide minerals are then dissolved. In the second stage, after aeration of the environment, Fe(II) oxidizes and reprecipitates, generally as Fe(III) (oxyhydr)oxide minerals under circumneutral pH conditions, forming films that become thicker during each cycle, potentially leading to lateritic duricrust formation [11].

Such iron cycling is not exclusively inorganic but also driven by microbial iron reduction [12,13]. Two bacterial metabolic pathways to dissolve iron, i.e., dissimilation or fermentation, can be distinguished [14]. For dissimilation, bacteria obtain energy for the respiratory process from carbohydrates and use Fe(III) as the final electron acceptor [15]. For that, mineral iron can be accessed by extracellular electron transfer, which requires direct contact of the cell with the Fe(III) mineral surface by the use of pili-like organic conductive structures (microbial nanowires) for distant electron transfer or indirectly through the solubilization of Fe(III) by organic chelators [16,17]. For fermentation, bacteria do not use Fe(III) as a final electron acceptor, but the mobilization of electrons during fermentation metabolism can stimulate the Fe(III) reduction process indirectly [18,19]. In both cases, the growth of iron-reducing microorganisms under favorable conditions can form biofilms, a matrix rich in extracellular polymeric substances [20]. The generation of this organic structure, along with iron precipitation under oxidizing conditions, may result in substrate aggregation and consolidation, potentially restoring lateritic crusts by biocementation [21].

To accelerate the biocementation in post-mining landscapes, the inoculation with iron-reducing bacteria may be necessary to restore *cangas*, especially in environments where such organisms are naturally scarce or lacking [22]. As iron reduction requires anaerobic conditions, effective iron-reducers are expected to inhabit aquatic ecosystems such as pools, lakes and crevices of natural iron-rich *canga* environments, often characterized by intense iron cycling [9]. Further iron-rich habitats, specifically those that present cycles of anaerobic and aerobic conditions promoted by flooding and drought regimes, may harbor candidate enrichment culture to induce the biocementation of substrates rich in iron oxides, enabling the large-scale restoration of lateritic crusts in the post-mining landscape.

In order to produce iron-reducing enrichment cultures promising to mediate *canga* duricrust restoration, we collected samples from pristine *canga* and anthropized environments from the Carajás National Forest, Eastern Amazon, Brazil. The samples were cultivated in a specific cultures medium containing Fe(III) citrate as an iron source that induces iron reduction by dissimilatory and fermentative processes. We compared the iron-reducing potential of the collected enrichment cultures using the Ferrozine test. We selected the enrichment cultures with the highest iron-reducing potential and transferred them to a growth medium in order to characterize their growth, cell type and electron transport system activity, which provides important information about the management and suitability for the application of the enrichment cultures in the field [23,24,25]. To test the large-scale viability of the selected enrichment cultures, we accompanied for four months their iron reduction using crushed duricrusts as a unique iron source. Finally, we identified the composition of the selected enrichment cultures by sequencing 16S rDNA and analyzed the main metabolic functions associated with iron cycling by the enrichment cultures. By that, we were able to obtain two promising enrichment cultures, which present favorable growth patterns and suitable biological activity for laboratory and field biocementation tests aiming to restore *canga* environments.

## 2. Materials and Methods

### 2.1. Geographical Setting and Sampling

This study was carried out in the *cangas* and iron mines of the Serras de Carajás, located in Southeastern Amazon, Pará, Brazil (Figure 1). The region has a humid tropical climate with an annual rainfall close to 1800 mm, concentrated in a rainy season from November to March, and an average annual temperature of 26 °C [26]. The Carajás *cangas* are patchy ecosystems surrounded by different forest types on altitudes around 700 m above sea level, grouped within two distinct mountain ridges, *Serra Norte* (SN) and *Serra Sul* (SS). Emerging above lateritic crusts, the *cangas* comprise different habitats such as lakes and periodically inundated areas [27]. Previous chemical data revealed iron cycling in the studied permanent and temporary lakes, evidenced by dissolved Fe concentrations up to 0.01 mmol [28].

To obtain iron-reducing enrichment cultures, sediments and water samples were collected in a small pond of the lateritic duricrust (1-SS), at the margins of a small creek (2-SS), at the margins of permanent lakes (3-SS, 5-SS), in a periodically inundated grassland (4-SS), a *canga* cover-soil distributed on top of an iron mining waste pile (6-SN), and a temporary lake (7-SN) (Figure 1, Appendix A). All sampling points were georeferenced using a handheld Garmin GPS device. Water samples were collected using one-way pipettes, and sediments were sampled using disposable spatulas. Samples were stored in sterile falcon tubes (50 mL) and kept under anoxic and refrigerated conditions until arrival in the laboratory. To guarantee anoxic conditions in falcon tubes containing sediments, we used AnaeroGen Compact paper sachets (Oxoid LTD, Hampshire, UK).

### 2.2. Enrichment, Selection and Propagation of Microbial Cultures

To promote the selective growth of iron-reducing microorganisms, 5 g (sediments) or 5 mL (liquids) from each sample were transferred to Schott Duran laboratory bottles containing 200 mL of DSMZ 579 culture medium. This medium contains 13.7 g/L Fe(III) citrate as an iron source and 2.5 g/L Na-acetate as carbon and energy source. For mineral and vitamin supply, the culture medium was enriched with 10 mL/L DSMZ 141 medium [10]. The pH was adjusted to 6.8 using 1 M H_2_SO_4_, and remaining oxygen was replaced by flushing N_2_ in the bottle. The samples were stored in a shaker chamber at a temperature of 28 °C and 163 rpm until further processing.

To select the enrichment cultures with the highest iron reducing potential, we transferred an aliquot of 50 mL of each inoculum to a new container with fresh culture medium (DSMZ 579). After seven days of cultivation, the soluble Fe(II) was quantified in triplicate (see Section 2.3). The two enrichment cultures with the highest concentrations of dissolved Fe(II) were transferred in DSMZ 579 medium, from which we substituted the carbon source Na-acetate by 3.36 g/L glucose. After completing seven days, the reduction of iron was again evaluated in triplicate. From these cultures, we took aliquots to measure bacterial growth (Section 2.4) and the activity of the electron transport system (ETSA, Section 2.5).

To outline the ability to reduce crystalline iron oxide, we cultivated selected enrichment cultures using crushed *canga* composed of hematite, goethite, magnetite and gibbsite (Appendix A) as iron source in a bioreactor. For that, we prepared a 1 L Schott bottle containing 800 mL of DSMZ 579 culture medium enriched with 3.36 g/L of glucose. The volume was completed with a mix of 4-SS and 6-SN inoculum in equal parts, and was kept in a shaking chamber at 28 °C and 163 rpm for four months. Iron reduction was evaluated monthly using the ferrozine test.

### 2.3. Ferrozine Assays

The potential for Fe(II) production of the enrichment cultures was detected spectrophotometrically using the ferrozine assay [29]. For that, 0.1 mL enrichment culture was mixed with 5 mL of 0.5 M HCl for 15 min. Then, an aliquot of 0.2 mL was added to 2 mL of ferrozine (1 g/L in 50 mM HEPES). As control, we used culture medium instead of enriched cultures. For bioreactor assays, we used ultrapure water and raw culture medium kept under the same conditions. After one hour, the sample was filtered (0.2 μm pores) to remove interferents, and the absorbance reading was performed at 562 nm (EZ Read 2000-Biochrom). The Fe(II) concentration was estimated by comparison of detected absorbance with the standard absorbance curve for ferrous ethylenediammonium sulfate and expressed in micromole (mmol) [29]. To check for differences in iron reducing potential among enrichment cultures and control treatments, data were checked for normal distribution, and one-way analysis of variance (ANOVA) was carried out, followed by post-hoc Tukey tests. Statistical analyses were performed in R Environment [30].

### 2.4. Microbial Growth and Microscopy

To describe the growth pattern of the selected enrichment cultures, we counted cells in a standardized volume every 24 h for three days. Seven days after the second ferrozine test, selected samples were enrichment cultivated in DSMZ 579 medium with glucose. After each interval of 24 h (0, 24, 48 and 72 h), an aliquot of 0.5 mL was fixed in sterile 4% formaldehyde for cytometric analysis (FACS Aria II, BD Biosciences, San Jose, CA, USA). Sample preparation was adapted from [31], where 500 µL of the fixed samples are added in sterile tubes with 2500 µL of sterile saline solution (pH = 7) and homogenized by Vortex Mixer VX-200 (Labnet International, Edison, NJ, USA). Then, the samples were centrifuged at 3000 rpm for 10 min at 4 °C, with the supernatant (1500 µL) collected and filtered in antistatic tubes using a 0.2 µm pore filter from Millipore Isopore (Merck, Tullagreen, Ireland), to remove debris. 200 µL of the filtered material had the volume equalized and received 400 µL of saline solution. For final reading, an aliquot of 450 µL was used.

Cytometric counting was performed by UV laser excitation on bacterial cells stained with SYBR-Green I (Molecular Probes, Invitrogen, Karlsruhe, Germany) under the 398 mV forward scatter and 972 mV side scatter settings. Calibration of counts was performed using standardized beads (Cytomics FC 500 CXP, Beckman Coulter, Krefeld, Germany) and the number of cells was estimated according to the formula described in [32]. From the number of cells, the number of generations, generation time and growth rates of cultures were calculated following Murray et al. [33].

The morphology of the selected enrichment cultures agents was observed under a microscope. For that, the samples were stained using a 4,6-diamidino-2-phenylindole (DAPI) at a concentration of 10 mg/mL^−1^ [34]. Following this, 100 µL of the material was filtered through a Millipore Isopore membrane (0.22 m), which was then mounted on a microscope slide for direct inspection. Observations were performed at 100× magnification using the Axio Imager M2 microscope (Carl Zeiss GmbH, Oberkochen, Germany) equipped with a fluorescent light source (HXP 120 C metal halide lamp). The fluorescence spectral lines ranged from 359–579 nm, with an excitation range of 335–383 and an emission range of 420–470 nm. DAPI dye has an excitation of 359 nm and an emission of 461 nm.

### 2.5. Activity of the Electron Transport System (ETSA)

To outline metabolic activity of the selected enrichment cultures during the exponential growth phase, the electron transport system activity assay, adapted from [35], was carried out. During the assay, INT (2-[(piodophenyl)-3-(p-nitrophenyl)-5-phenyl tetrazolium chloride]) competes for electrons with NAD and FAD, being reduced by components of the electron transport chain in metabolically active cells. The reduction of INT forms INT-Formazan (iodonitrotetrazolium chloride), which can be detected spectrophotometrically. In this way, it provides an estimate of microbial metabolism by evaluating the electron transport system activity (ETSA) [36].

To analyze the ETSA, a volume of 1 mL of inoculum and 0.2 mL of 8 mM INT was added to Eppendorf Tubes^®^ with caps. As control, we used Eppendorf tubes containing 1 mL of the enrichment culture and 0.2 mL of deionized water. The tubes were incubated for 35 min in the dark. Then, 5 mL of methanol was added and the samples were centrifuged (10 min at 3000 rpm). The supernatant was skimmed and absorbance was measured at a wavelength of 475 nm (EZ Read 2000-Biochrom, Biochrom Ltd., Cambourne, Cambridge, UK).

Statistical differences between control treatment and ETSA assay were detected using Student *t*-tests after checking data for normal distribution.

### 2.6. Molecular Analysis

In order to characterize community composition of selected enrichment cultures, we performed DNA sequencing in triplicate. Approximately 15 mL of each enrichment culture were filtered through sterile 0.22 µm filter membranes (Millipore, Merck KGaA, Darmstadt, Germany). For extraction, the membranes were manually cut with a sterile scalpel into small pieces and DNA was extracted using the PowerSoil DNA Isolation Kit (QIAGEN, Hilden, Germany) according to the manufacturer’s instructions. DNA samples were quantified using Qubit 3.0 fluorometer (Thermo Fisher Scientific Inc., Waltham, MA, USA) and DNA quality was verified in a 1% electrophoresis agarose gel (100 V).

Approximately 5 ng μL^−1^ of extracted DNA from each sample were used for PCR reaction. The gene-specific sequences used target V3–V4 region of the 16S rRNA gene [37]. Illumina adapter overhang nucleotide sequences were added and this region was amplified using the full-length primer set 16S Forward Primer = 5′ TCGTCGGCAGCGTCAGATGTGTATAAGAGACAGCCTACGGGNGGCWGCAG 3′ and 16S Reverse Primer = 5′ GTCTCGTGGGCTCGGAGATGTGTATAAGAGACAGGACTACHVGGGTATCTAATCC 3′. The PCR was prepared in a final volume of 25 μL completed with ultrapure water. The mix contained 1.25 μL of dNTP (2 mM), 2,0 μL of MgCl_2_ (25 mM), 0.5 μL of each primer (10 μM), 1μL of DNA (5 ng), 1 μL of Platinum™ Taq DNA polymerase (Invitrogen, Waltham, MA, USA), and 5 μL 10× Buffer. Amplification was performed in a thermocycler Applied Biosystems with an initial cycle of 3 min at 95 °C, followed by 25 cycles of 30 s at 95 °C, 30 s at 57 °C, and 30 s at 72 °C, with a final cycle of 5 min at 72 °C.

The amplicon libraries for the bacteria were prepared according to the Illumina 16S Metagenomic Sequencing Library Preparation Protocol (Illumina, San Diego, CA, USA). We applied a paired-end sequencing technology (MiSeq Illumina) after DNA extraction, purification and amplification to build metagenomic libraries. The Index PCR mix contained 2.5 μL DNA from the previous PCR, 12.5 μL of 2× Kapa Hifi HotStart Ready Mix (Sigma-Aldrich, St Louis, MI, USA) and 10 μL of Index primer (1 μM). The PCR cycle for bacteria consisted of an initial denaturing of 3 min at 95 °C, followed by 25 cycles of denaturation at 95 °C for 30 s, annealing at 57 °C for 30 s, extension at 72 °C for 30 s, and a final extension at 72 °C for 5 min. The concentrations of the PCR fragments were measured in a Qubit fluorometer using a Qubit™ ds DNA HS Assay (Thermo Fisher Scientific, Waltham, MA, USA). The size and quality of the PCR fragments were estimated on an Agilent 2100 Bioanalyzer (Agilent Technologies, Santa Clara, CA, USA) using a DNA 1000 chip. The PCR reactions were purified with an AMPure XP purification kit (Beckman Coulter, Brea, CA, USA), and the libraries were further processed with a Nextera XT kit (Illumina). The libraries were standardized to a concentration of 4 nM and processed following Illumina 16S Metagenomic Sequencing Library Preparation (Illumina). The 16S rRNA gene libraries were sequenced in a Miseq-Illumina platform using a MiSeq V3 reagent kit (600 cycles; Illumina).

The 16S rRNA sequences were analyzed following the PIMBA pipeline (Pipeline for MetaBarcoding Analysis) [38]. The sequences were trimmed and filtered by quality and converted to FASTA using Prinseq v0.20.4. VSEARCH v2.15.2 was performed to dereplicate, discard singletons, trim reads and group sequences into operational taxonomic units (OTUs) with 97% similarity. Chimeras were removed. Taxonomic assignment was performed using the SILVA ribosomal RNA databases, SSU 132 [39]. For functional characterization of the detected taxa, we applied the Functional Annotation of Prokaryotic Taxa prediction tool (FAPROTAX, version 1.2.4). FAPROTAX is a database to map metabolic functions of bacterial OTUs into putative functional profiles, enabling ecological interpretation of 16S marker gene data. The database distinguishes 80 functions within more than 7600 functional annotations covering 4600 taxa; analysis were performed using collapse_table.py script [40].

## 3. Results

### 3.1. Iron Reduction of Enrichment Cultures

Seven days after incubation in DSMZ 579 enriched with Fe(III) citrate, six out of the seven enrichment cultures can be considered Fe(III) reducers. Regarding their origin, four iron reducing enrichment cultures were obtained in Serra Sul, and two were from Serra Norte (Figure 2). A single sample from Serra Sul did not reduce Fe(III). The 4-SS and 6-SN enrichment cultures showed the highest Fe(II) concentrations, three or more times higher than the rest of the samples, and were selected as the most promising Fe(III)-reducing cultures. 4-SS was from a periodically flooded grassland, while 6-SN was obtained from an iron mining waste pile.

After feeding with glucose (containing Fe(III) citrate as iron source), both enrichment cultures increased their Fe(II)-reducing activity (Figure 2). Iron reduction in the bioreactor, containing a mix of both enrichment cultures and crushed *canga* as the iron source, shows exponential growth (Figure 3). Upon 30 and 60 days after incubation, iron reduction was detectable but lower than in cultures fed with Fe(III) citrate and glucose. The concentration of reduced iron increases and reaches 1.78 mmol/L of dissolved iron at 120 days.

### 3.2. Activity of the Electron Transport System

After seven days in the growth medium, the activity of the electron transport system of the 4-SS and 6-SN Fe(III)-reducing cultures was significantly higher than the control (Figure 4).

### 3.3. Microbial Growth and Microscopy

The highest microbial density of enrichment cultures was observed at 48 h after transfer to growth medium (Figure 5A,B, Appendix A). At this time, the 6-SN enrichment culture showed 1.4 times more viable cells than 4-SS. Regarding the morphology, there was a predominance of rod-shaped cells in both enrichment cultures (Figure 5C,D). In the 4-SS enrichment culture, the average cell length was smaller (1.5 μm) than in the 6-SN enrichment culture (1.8 μm).

### 3.4. Microbial Composition

Sequencing revealed a total of 420,803 sequences of 93 OTUs in the two selected enrichment cultures (Appendix A). All OTUs were identified down to the family level, and 27 were classified at the genus level. Fourteen OTUs were present only in the 4-SS enrichment culture, sixty-six only in the 6-SN enrichment culture, and thirteen OTUs were shared (Appendix A).

The two families Rhizobiaceae and Burkholderiaceae were frequent in 4-SS. The Rhizobiaceae family incorporated 56% of all OTUs detected in this enrichment culture (Figure 6). 57% of the OTUs found in this enrichment culture are aerobic bacteria; noteworthy are the genera *Nitratireductor* and *Ensifer*. In the 6-SN enrichment culture, the three families Enterobacteriaceae, Burkholderiaceae and Paenibacillaceae were identified, which correspond to facultative anaerobic, Fe-reducing and/or Fe-oxidizing bacteria. The Enterobacteriaceae family dominated the enrichment culture with 51 OTUs of the genus *Serratia* and six OTUs from *Enterobacter*, while the genus *Acidovorax* represented the Burkholderiaceae family. The Paenibacillaceae family had only one representative, *Paenibacillus* sp. (Appendix A). The OTUs shared by both enrichment cultures belong to the families Rhizobiaceae, Burkholderiaceae and Enterobacteriaceae. At the genus level, *Achromobacter* spp. and *Serratia* sp. were dominant.

The functional annotations of the identified OTUs indicated differences between 4-SS and 6-SN enrichment cultures. It was possible to detect function annotations related to respiratory activity in the 4-SS enrichment culture, while, in the 6-SS enrichment culture, fermentative processes dominated functional annotations. Chemoheterotrophy and nitrate-reducing activity were identified in both enrichment cultures (Figure 6).

## 4. Discussion

In this study, it was possible to obtain iron-reducing enrichment cultures from different environments. Based on their iron-reducing potential, we selected two promising enrichment cultures able to reduce iron from different sources, including a natural, iron-rich substrate. The best iron-reducing enrichment cultures were gathered from distinct habitats, i.e., a natural, temporarily flooded grassland and a post-mining environment, and differ regarding composition and functional mechanisms to reduce iron. As iron-reducing bacteria are widely distributed, even in post-mining environments, the inoculation of enrichment cultures during to trigger biocementation in the field may be dispensable. However, the selected enrichment shows favorable metabolic activity and growth parameters for large-scale applications during post-mining biocementation, and their potential for duricrust restoration should be tested in the field.

Applying Fe(III) citrate as an iron source in the medium successfully promoted microbial iron-reduction. It resulted in the enrichment of obligate or facultative iron reducers and effective production of enrichment cultures. The two chosen enrichment cultures removed more iron than was naturally found in the water bodies of the lakes from the *cangas* of Carajás [28], demonstrating how glucose, as opposed to Na-acetate, stimulates bacterial activity. Both enrichment cultures show (i) high ETSA activity, indicating a high metabolism rate, which leads to the production of organic acid compounds, facilitating Fe(III) solubilization [41] and (ii) growth patterns favorable for their conditioning and multiplication in bioreactors. The chosen enrichment cultures were able to reduce significant amounts of iron from crushed, iron-rich rocks when housed in bioreactors, demonstrating their potential to scale up iron reduction for biocementing post-mining landscapes.

The most promising enrichment cultures were retrieved from a periodically flooded grassland (4-SS) and on the surface of an iron mining waste pile (6-SN). The presence of effective iron reducers in these environments may result from repeated cycles of flooding/soaking and desiccation throughout the year that both locations undergo, especially in the transition periods between the wet and dry seasons. This should favor iron-reduction and may contribute to the colonization of iron-reducing biota in these locations. Previous studies have shown that redox fluctuations drive the repeated dissolution and precipitation of Fe(III) minerals, influencing the degree of crystallinity of minerals and the bioavailability of Fe(III) [42]. This way, soils or rocks exposed to rainfall events that produce conditions of iron reduction and a periodic accumulation of Fe(II) can be prepared for faster rates of iron reduction than those characterized by extensive droughts or continuously wet systems such as permanent lakes [43].

Obtaining iron-reducing enrichment cultures from different types of environments confirms the wide distribution of iron-reducing microorganisms in nature, although differences between samples have been found. This has two implications for large-scale biocementation in practice. First, the wide distribution of iron-reducing bacteria makes it possible to obtain local enrichment cultures (Serra Sul and Serra Norte are separated by about 40 km) without the need to translocate enrichment cultures to induce biocementation in the field, reducing the risks associated with the introduction and translocation of biota between areas [44]. Our data indicate that environments with different cycles of flooding/soaking and dissection are most promising habitats to gain iron-reducing enrichment cultures. Second, the presence of iron-reducing bacteria in an iron mining waste pile may make the enrichment cultures with these organisms dispensable, as solely the management of the present microbial community may promote iron reduction and duricrust restoration at this location.

Both enrichment cultures obtained in this study showed high rates of iron reduction, but 6-SN enrichment culture surpassed 4-SS in terms of the number of OTUs (diversity), cell density, ETSA activity, and reduced iron, which may result from differences in composition. The genus Serratia, capable of coupling the oxidation of glucose or acetate with the reduction of iron to conserve energy for growth [45,46], was detected in both enrichment cultures, but Alpha-, Beta-, Delta- and Gammaproteobacteria classes dominated in the 4-SS enrichment culture. Although commonly found in flooded regions, including Carajás [47], they break down complexes of organic compounds but do not reduce iron. In the 6-SN enrichment culture, some genera associated with microbial iron reduction that were not found in 4-SS, such as a strain of *Paenibacillus* and OTUs associated with the Enterobacter genus, stand out. In the literature, some species of *Paenibacillus* are described to perform the dissimilatory reduction of iron, showing the expression of genes involved in the synthesis of siderophores, and also in the release of ferrireductase [48,49]. The Enterobacter genus is furthermore known for the fermentative reduction of Fe(III), especially in more crystalline Fe oxides [14,50]. Overall, higher diversity, the presence of different functional annotations, and superior cell density in 6-SN when enriched with glucose may have contributed to the higher Fe reduction and better performance of ETSA in the 6-SN enrichment culture.

## 5. Conclusions

Here, we presented a method to successfully extract and propagate local iron reducing enrichment cultures from field samples comprising different functional groups, geographical spaces and habitats. Given the distribution and abundance of iron-reducing microorganisms, the method enabled the generation of local enrichment cultures without the need to transfer non-native microorganisms to *canga* restoration sites.

When successfully applied for duricrust restoration, these enrichment cultures may benefit the endangered biodiversity associated with these rare environments and contribute to more sustainable iron mining, and their potential for bioremediation in post-mining landscapes should be tested under controlled and field conditions. Laboratory assays and field experiments to outline the benefits of enrichment cultures in comparison to medium or water only treatments for biocementation of iron mining waste are underway.

## Figures and Tables

**Figure 1 microorganisms-11-00448-f001:**
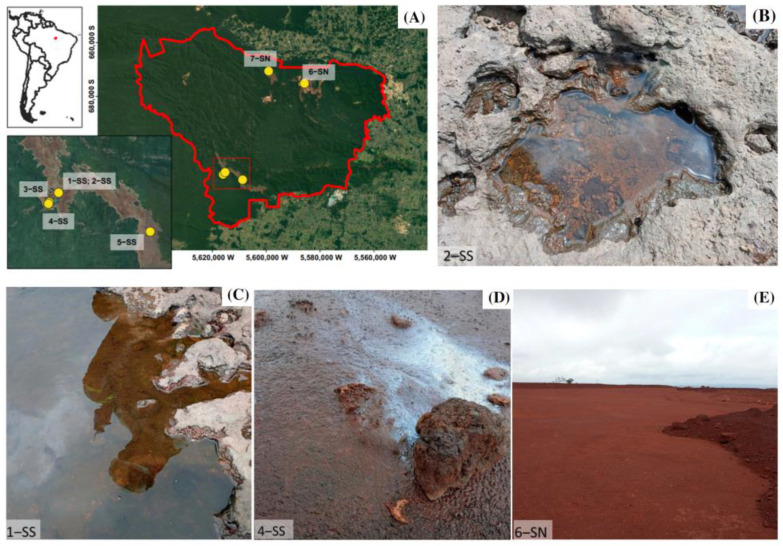
Location of sampling points in the Carajás National Forest (red line, **A**), and details of the selected sampling locations (**B**–**E**).

**Figure 2 microorganisms-11-00448-f002:**
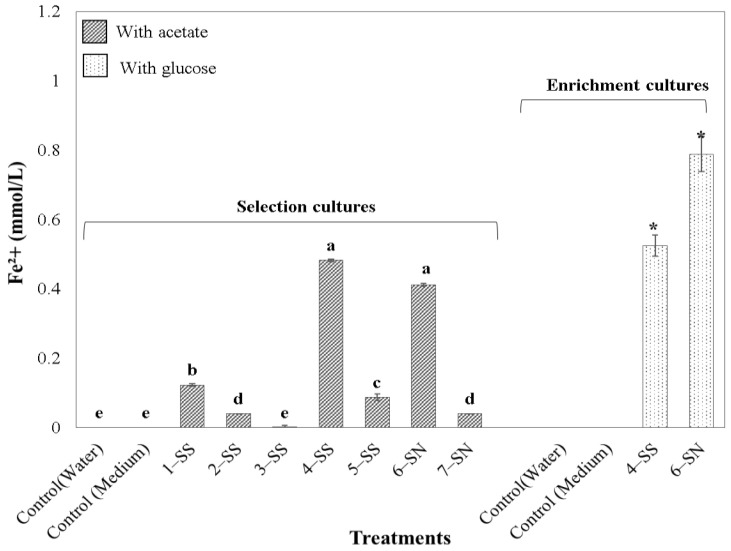
Iron reduction measured as Fe(II) concentration of enrichment culture obtained from field samples from the Serra dos Carajás, Brazil. On the left, samples were incubated with Na acetate for seven days; on the right, iron reduction after feeding with glucose for seven days. Different letters above the bars indicate significant differences according to a Tukey post hoc test (*p* < 0.05). Asterisks indicate significant differences between enrichment cultures grown in acetate and glucose medium.

**Figure 3 microorganisms-11-00448-f003:**
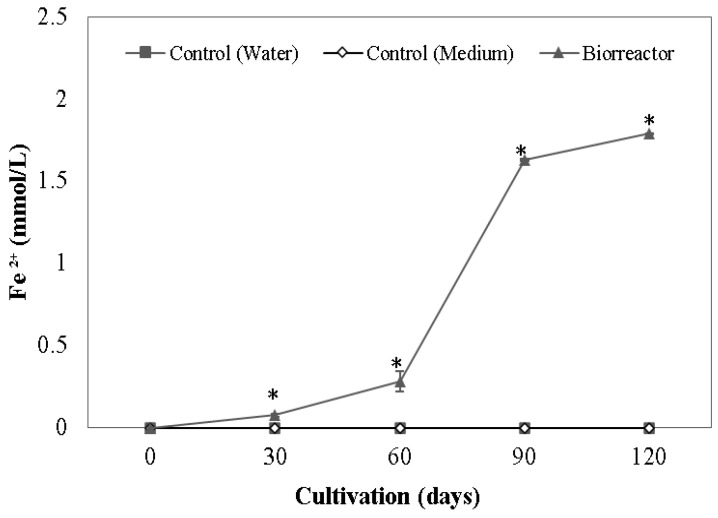
Concentration of dissolved Fe(II) in the bioreactor containing a mix of enrichment cultures 4-SS and 6-SN. (*) Means differ significantly from the control treatments by the Student’s *t*-test (*p* > 0.05).

**Figure 4 microorganisms-11-00448-f004:**
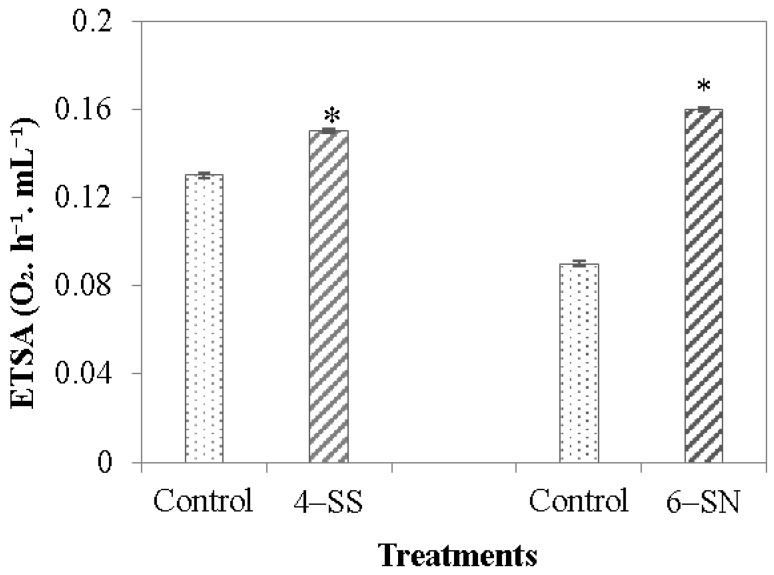
Activity of the respiration electron transport system (ETSA) of 4-SS and 6-SN enrichment cultures after seven days of cultivation in a growth medium. (*) Means differ significantly from the control by the Student’s *t*-test (*p* > 0.05).

**Figure 5 microorganisms-11-00448-f005:**
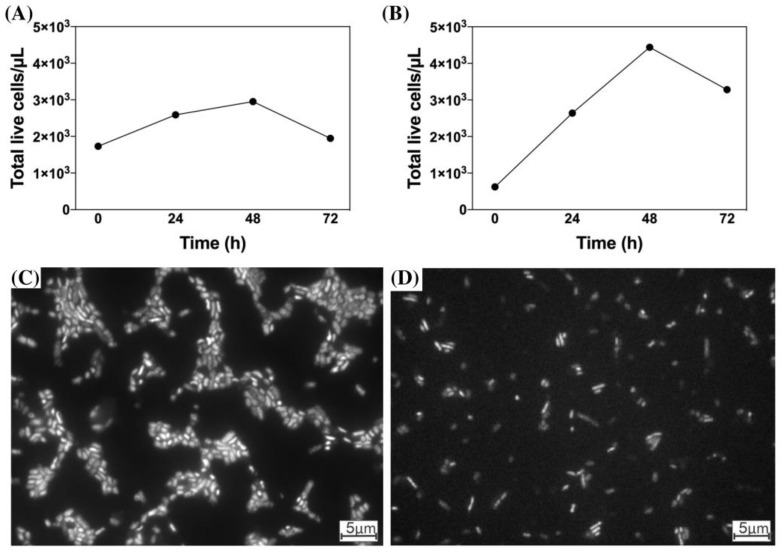
Cell growth (number of cells, **A**,**B**) and microscopic aspects (**C**,**D**) of selected enrichment cultures 4-SS (**A**) and 6-SN (**B**) in culture medium enriched with glucose. Microscopic photographs at 100× magnification using fluorescence microscopy.

**Figure 6 microorganisms-11-00448-f006:**
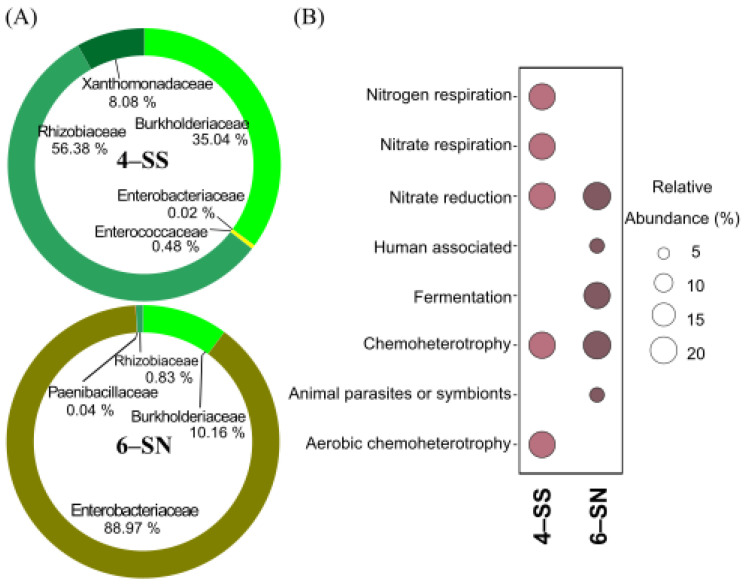
Composition (**A**) and functional characterization (**B**) of selected enrichment cultures 4-SS and 6-SN.

## Data Availability

Not applicable.

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
