# Peer review of "Acquiring Iron-Reducing Enrichment Cultures: Environments, Methods and Quality Assessments"

_microorganisms, 2023, doi:10.3390/microorganisms11020448_

Round 1

Reviewer 1 Report

1. Please state the specific collection environment for samples of 3-ss and 5-ss.

2. Whether a parallel test was set in the iron reduction potential assays for inocula obtained from field samples, and if not, is there a accidental error in the results

3. In the ETSA assay, why the control group with deionized water has a higher electron transport system activity, and the results of the two control groups were inconsistent? In addition, the gap between the two control groups is larger than that between sample 4-ss and the control group. Can it be explained that the error of the test results is large?

4. In line 131-133, “For that, data were checked for normal distribution, and one-way analysis of variance (ANOVA) was carried out, followed by post-hoc Tukey tests. Statistical analyses were performed in R Environment [32]”, how is this reflected in the paper?

5. In line 160-162, “From the number of cells, the number of generations, generation time and growth rates of cultures were calculated following [36].” No relevant data in the paper.

6There is not enough data to support the discussion. Such as there is no detailed description of the formation environment of the sample collection site in the paper, but the conclusion is derived directly from the different collection environment in the discussion. And the data in this paper lack comparison of different media for glucose and Na-acetate.

Author Response

  1. Please state the specific collection environment for samples of 3-ss and 5-ss.

Thanks for suggestion. We added the required information in the method section (l. 103-107) and placed geographic coordinates and an additional habitat description of the collected inocula in Supplementary Material 1.

  1. Whether a parallel test was set in the iron reduction potential assays for inocula obtained from field samples, and if not, is there a accidental error in the results

Thank you for the important comment. We did not carry out a parallel test of iron reduction potential, but we exclude the possibility that accidental errors influence the results, as we tested each inocula measurement against a control probe containing water and medium only).

  1. In the ETSA assay, why the control group with deionized water has a higher electron transport system activity, and the results of the two control groups were inconsistent? In addition, the gap between the two control groups is larger than that between sample 4-ss and the control group. Can it be explained that the error of the test results is large?

Thanks for your suggestion, we added details of the ETSA assay in the method section that explain the low differences.

  1. In line 131-133, “For that, data were checked for normal distribution, and one-way analysis of variance (ANOVA) was carried out, followed by post-hoc Tukey tests. Statistical analyses were performed in R Environment [32]”, how is this reflected in the paper?

Sorry for the oversight here, we performed Student T-tests to check for differences between control and treatments (ETSA assays, l. 184), and ANOVA to test for differences between treatments (ferrozine assays, l. 150-153). Results of statistical tests are embedded in Figures 2 and 3.

  1. In line 160-162, “From the number of cells, the number of generations, generation time and growth rates of cultures were calculated following [36].” No relevant data in the paper.

As these data are of minor importance for the objective of the manuscript, this information is placed in supplementary material in the revised version. Please check table “Table S2”.

6There is not enough data to support the discussion. Such as there is no detailed description of the formation environment of the sample collection site in the paper, but the conclusion is derived directly from the different collection environment in the discussion. And the data in this paper lack comparison of different media for glucose and Na-acetate

Thanks for observation. We added detailed descriptions of the environments where we sampled the inocula (l. 103-107) to support discussion (l.328) and conclusions (l.391-392). We modified Fig. 2 (l.267) in order to make it more clear the comparison between selected enrichment cultures.

Reviewer 2 Report

Comments:

The manuscript Microrganisms-2065393 entitled “Acquiring iron-reducing microbial inocula: Environments, methods and quality assessments” presents an interesting and innovative methodology to restore lateritic crusts, canga environments, by microbiological techniques, which contribute to more sustainable iron mining in the Serra dos Carajás region. However, the manuscript need a good revision prior to its publication acceptance.

Specific Comments:

1.      The manuscript presents a relative good writing and English, but there are some minimal form mistakes.

2.      The Ferrozine assays was not totally clear.

3.      Concentration unit is mol/L not M or mM.

4.      The components of the inoculum did not affect the spectrophotometer analysis? The inoculum should present some turbidity, which may probably affect the spectrophotometric analysis.

5.      Normally, microorganism’s growth curves are plotted considering more than 4 points. In the present way is not possible to appreciate the typical phases of the growth.

6.      The reproducibility of the results in a higher scale was studied? Would it be possible? In order to culture in bioreactors (higher scale) knowledge of the microorganism’s growth curve (detailed) will be mandatory.

7.      How exactly was detected the function related to respiratory/fermenting activity? It was not clear on the text. It would be interesting a better exploration/discussion of this topic, since the determination of the reduction mechanism is one of the most relevant fundamental in these studies.

Author Response

  1. The manuscript presents a relative good writing and English, but there are some minimal form mistakes.

Thanks for the positive evaluation of our work! As explained in the following comments, we attained all suggestions and comments and throurouhly revised the manuscript.

  1. The Ferrozine assays was not totally clear.

We revised this part of the text and reformulated, adding details about control treatments and standard absorbance curves clearer. Furthermore, we revised the statistical treatment of the data. Please see revised Method section (l. 140-153).

  1. Concentration unit is mol/L not M or mM.

Thanks for suggestion, we standardized to mmol/L in methods (l. 99 and 150) and results (I. 265).

  1. The components of the inoculum did not affect the spectrophotometer analysis? The inoculum should present some turbidity, which may probably affect the spectrophotometric analysis.

The samples were filtered to reduce the influence of debris on measurement of light interaction on ferrozine. We added this essential information to the text (l. 146).

  1. Normally, microorganism’s growth curves are plotted considering more than 4 points. In the present way is not possible to appreciate the typical phases of the growth.

I appreciate your observation. As we choose to streamline the analysis procedures and integrate more information on cellular metabolism, our work consists of a few key points only. This allows to break down the growth curve into four phases: the lag phase, log phase, or exponential phase, the stationary phase, and the decline phase, or phase of cell death. We deduced from the counting data that the cells had a log phase within 48 hours and then started to decline, although more measurements would have permit to appreciate the limits between singles phases more exactly. Nevertheless, we consider our observations as sufficient to large-scale operations with the enrichment cultures, which we will shortly perform.

  1. The reproducibility of the results in a higher scale was studied? Would it be possible? In order to culture in bioreactors (higher scale) knowledge of the microorganism’s growth curve (detailed) will be mandatory.

We added information of the iron reducing potential of the enrichment cultures in a bioreactor, using crushed canga as iron source to illustrate the possibility to achieve higher scale. Please see l. 133-139 (Methods), l. 261-266 (Results) and l. 342 and 345 in the discussion section..

  1. How exactly was detected the function related to respiratory/fermenting activity? It was not clear on the text. It would be interesting a better exploration/discussion of this topic, since the determination of the reduction mechanism is one of the most relevant fundamental in these studies.

Thanks for the suggestion. We added a short description of the FACTOPRAX database and software used to make the functional annotations at the end of the method section to make the first topic clear. As we do not own direct evidence that the enrichment cultures in fact reduce iron by one or the other metabolic way, we toned down our conclusion in the revised manuscript, added literature on this issue and simplified our affirmation to highlighted that our findings are based on the assumption of different functional annotations (l. 388).

Reviewer 3 Report

Acquiring iron-reducing microbial inocula: Environments, methods and quality assessments, by Aline Cardoso, Rayara da Silva, Isabelle Prado, José Bitencourt, and Markus Gastauer

Submitted to Microorganisms

This work describes a method to obtain enrichment cultures of Fe-reducing microorganisms, and the results they obtained with this method. Results are interesting, but were not well explored. Instead of exploring the results, there are several mentions to the use of these enrichment cultures as inocula for environmental restoration, which was not described in the methods and results sections. It seems that the goal of the project is to use the enrichment cultures as inocula for environmental restoration; but the present work is about enrichment cultures of Fe-reducing microorganisms. In addition, there is a confusion between the inocula they collected to start the enrichment cultures, and the enrichment cultures which are called “inocula” throughout the text. Thus, I recommend that the text is largely revised to describe and discuss the actual results, which are about enrichment cultures, not inocula. I recommend the use of “enrichment culture(s)” instead of “inocula” and “inoculum” throughout the text. Future work should be mentioned at most once, at the end of the manuscript. To correct this issue, the manuscript should be largely re-written, specially the title, abstract, introduction and discussion sections.

The title does not reflect what was done and should be changed (see the reasoning above).

In the Methods section, description of methods used to maintain the enrichment cultures anoxic are restricted to the sample collection and transport.

Several affirmations were erroneously attributed to cited papers. Please correct them.

The usage of language is good most of the time, although in some points the text is rather confusing. I tried to help improving the text.

Specific issues were annotated directly in the pdf file of the manuscript.

Author Response

  1. This work describes a method to obtain enrichment cultures of Fe-reducing microorganisms, and the results they obtained with this method. Results are interesting, but were not well explored. Instead of exploring the results, there are several mentions to the use of these enrichment cultures as inocula for environmental restoration, which was not described in the methods and results sections. It seems that the goal of the project is to use the enrichment cultures as inocula for environmental restoration; but the present work is about enrichment cultures of Fe-reducing microorganisms.

In addition, there is a confusion between the inocula they collected to start the enrichment cultures, and the enrichment cultures which are called “inocula” throughout the text. Thus, I recommend that the text is largely revised to describe and discuss the actual results, which are about enrichment cultures, not inocula. I recommend the use of “enrichment culture(s)” instead of “inocula” and “inoculum” throughout the text. Future work should be mentioned at most once, at the end of the manuscript. To correct this issue, the manuscript should be largely re-written, specially the title, abstract, introduction and discussion sections.

Thanks for your recommendations. We revised the objective section as suggested and substitute the term inocula by enrichment culture. We detailed necessary follow-up research in the conclusion section.

  1. The title does not reflect what was done and should be changed (see the reasoning above).

We accepted your suggestion by modifying the title of the manuscript to “Acquiring iron-reducing enrichment culture: Environments, methods and quality assessments”

  1. In the Methods section, description of methods used to maintain the enrichment cultures anoxic are restricted to the sample collection and transport.

We added more details about it in the item Methods (l.112). Culture medium preparation included the use of N2 to preserve the anoxic conditions.

  1. Several affirmations were erroneously attributed to cited papers. Please correct them.

We carefully checked all reviewer’s comments and suggestion in the attached pdf, and detail all actions to improve our manuscript below.

  1. The usage of language is good most of the time, although in some points the text is rather confusing. I tried to help improving the text.

Thanks for your support, your edits improved the quality of the text significantly.

  1. Specific issues were annotated directly in the pdf file of the manuscript.

We revised the manuscript considering all suggestions from the pdf file. Changes made in the revised version are highlighted in blue. We appreciate your valuable suggestions, which strongly contributed to the improvement of the manuscript. Please let us know where we can further improve our manuscript!

Reviewer 4 Report

This study aims to retrieve Fe(III)-reducing microbial cultures suitable to mediate restoration of canga environment. Although such microbial cultures may be useful, there are many uncertainness and scientific flaws, bringing bout serious damage to the current manuscript. The results obtained from only limited experiments do not fairly support the author’s conclusions.

1. It is questionable that the two cultures show high rates of Fe(III) reduction. Figure 2 shows the appearance of 0.4-0.8 mM Fe(II) in the cultures; such iron concentrations seem to be only small portions of initially added Fe(III) (as 13.7 g/L Fe(III)-citrate).

2. Authors used soluble Fe(III)-chelate for cultures. I think this makes the evaluation of Fe-reducing potential very difficult. Considering the application in canga restoration, they should use insoluble Fe(III) hydroxides such as goethite and ferrihydrite.

3. Figure 3 shows that ETSA values of control mixtures (deionized water + INT) seem high and comparable to those of microbial cultures. We can also interpret the result as the microbial cultures show only minor levels of ETSA.

4. Authors state “ highest growth rate was observed at 48 h” (Fig. 4). It is not rate but density or amount. What does the result of Figure 3 demonstrate? I do not think it shows the properties of microbial cultures with high ability to reduce Fe(III).

5. Authors discuss about the metabolic activities of selected two cultures based on the functional annotation of metagenome sequencing data (Fig. 5). However, it is only speculation. They should show direct evidence that the microbial cultures reduce Fe(III) via the respiratory or fermentation.

Author Response

This study aims to retrieve Fe(III)-reducing microbial cultures suitable to mediate restoration of canga environment. Although such microbial cultures may be useful, there are many uncertainness and scientific flaws, bringing bout serious damage to the current manuscript. The results obtained from only limited experiments do not fairly support the author’s conclusions.

Thanks foy your suggestions. We made major changes in the text and hope to have met your expectations.

  1. It is questionable that the two cultures show high rates of Fe(III) reduction. Figure 2 shows the appearance of 0.4-0.8 mM Fe(II) in the cultures; such iron concentrations seem to be only small portions of initially added Fe(III) (as 13.7 g/L Fe(III)-citrate).

To attribute reduced iron to the bacterial activity, we compared the results with the control treatment (Fig. 2). We agree with the reviewer that Fe(II) concentrations are low, and show in the added bioreactor experiment, that higher Fe(II) concentrations (up to 1.78 mmol/L after 120 days) require longer observational intervals than was available in the selection intervals.

  1. Authors used soluble Fe(III)-chelate for cultures. I think this makes the evaluation of Fe-reducing potential very difficult. Considering the application in canga restoration, they should use insoluble Fe(III) hydroxides such as goethite and ferrihydrite.

To address this issue, we added to the manuscript a second stage of our study, which included bioreactor preparation using canga itself as iron source. Thus, we were able to show that the cultures are effective iron reducers. Please see l. 133-139 (Methods), l. 261-266 (Results) and l. 342 and 345 in the discussion section..

  1. Figure 3 shows that ETSA values of control mixtures (deionized water + INT) seem high and comparable to those of microbial cultures. We can also interpret the result as the microbial cultures show only minor levels of ETSA.

Dear reviewer, we are sorry for a small oversight in the previous version of the manuscript regarding the control treatment in this assay. This explains the low difference.

  1. Authors state “ highest growth rate was observed at 48 h” (Fig. 4). It is not rate but density or amount. What does the result of Figure 3 demonstrate? I do not think it shows the properties of microbial cultures with high ability to reduce Fe(III).

Thanks for the suggestion the sentence has been rewritten in line 285: “The highest microbial density of enrichment culture was observed at 48h after transfer to growth medium ”

  1. Authors discuss about the metabolic activities of selected two cultures based on the functional annotation of metagenome sequencing data (Fig. 5). However, it is only speculation. They should show direct evidence that the microbial cultures reduce Fe(III) via the respiratory or fermentation.

Thanks for the observation. As we do not own direct evidence that the enrichment cultures in fact reduce iron by one or the other metabolic way, we toned down our conclusion in the revised manuscript, added literature on this issue and simplified our affirmation to highlighted that our findings are based on the assumption of different functional annotations (l. 388).

Round 2

Reviewer 4 Report

I thank the authors for their many efforts. Authors have revised the manuscript extensively, and this sounds to be fairly accepted by the Microorganisms readers.